# The Influence of Bowel Preparation on ADC Measurements: Comparison between Conventional DWI and DWIBS Sequences

**DOI:** 10.3390/medicina55070394

**Published:** 2019-07-21

**Authors:** Ilze Apine, Monta Baduna, Reinis Pitura, Juris Pokrotnieks, Gaida Krumina

**Affiliations:** 1Children Clinical University Hospital, LV-1004 Riga, Latvia; 2Department of Diagnostic Radiology, Riga Stradin’s University, LV-1038 Riga, Latvia; 3Faculty of Medicine, Riga Stradin’s University, LV-1007 Riga, Latvia; 4Department of Internal Diseases, Riga Stradin’s University, LV-1038 Riga, Latvia

**Keywords:** MR enterography, MRE, diffusion-weighted imaging, DWI, DWIBS, ADC

## Abstract

*Background and objectives:* The aim of the study was to assess whether there were differences between apparent diffusion coefficient (ADC) values of diffusion-weighted imaging (DWI) and diffusion-weighted imaging with background body signal suppression (DWIBS) sequences in non-prepared and prepared bowels before and after preparation with an enteric hyperosmolar agent, to assess whether ADC measurements have the potential to avoid bowel preparation and whether ADC-DWIBS has advantages over ADC-DWI. *Materials and Methods:* 106 adult patients without evidence of inflammatory bowel disease (IBD) underwent magnetic resonance (MR) enterography before and after bowel preparation. ADC-DWI and ADC-DWIBS values were measured in the intestinal and colonic walls demonstrating high signal intensity (SI) at DWI tracking images of b = 800 s/mm^2^ before and after preparation. *Results:* There were significant difference (*p* < 0.0001) in both ADC-DWI and ADC-DWIBS results between non-prepared and prepared jejunum for DWI being 1.09 × 10^−3^ mm^2^/s and 1.76 × 10^−3^ mm^2^/s, respectively, and for DWIBS being 0.91 × 10^−3^ mm^2^/s and 1.75 × 10^−3^ mm^2^/s, respectively. Both ADC-DWI and DWIBS also showed significant difference between non-prepared and prepared colon (*p* < 0.0001), with DWI values 1.41 × 10^−3^ mm^2^/s and 2.13 × 10^−3^ mm^2^/s, and DWIBS—1.01 × 10^−3^ mm^2^/s and 2.04 × 10^−3^ mm^2^/s, respectively. No significant difference between ADC-DWI and ADC-DWIBS was found in prepared jejunum (*p* = 0.84) and prepared colon (*p* = 0.58), whereas a significant difference was found in non-prepared jejunum and non-prepared colon (*p* = 0.0001 in both samples). *Conclusions:* ADC between DWI and DWIBS does not differ in prepared bowel walls but demonstrates a difference in non-prepared bowel. ADC in non-prepared bowel is lower than in prepared bowel and possible overlap with the ADC range of IBD is possible in non-prepared bowel. ADC-DWIBS has no advantage over ADC-DWI in regard to IBD assessment.

## 1. Introduction

Crohn’s disease (CD) is a chronic inflammatory bowel disease (IBD) of rising incidence and prevalence [1]. It has high complication rates [2], requiring surgical treatment upon progression [3] and resulting in a negative impact on patients’ quality of life [4]. It can however be treated more successfully if the disease is detected early.

Superior soft tissue contrast resolution enables magnetic resonance imaging (MRI) to track bowel inflammation beyond the reach of the endoscope. However, magnetic resonance enterography (MRE) requires bowel distension by an oral hyperosmolar enteric contrast agent [5], causing adverse effects sometimes leading to poor tolerance by patients [6,7]. MRE is contraindicated in patients requiring general anesthesia. MRE in IBD requires the use of intravenous gadolinium contrast agents [5], potentially causing systemic nephrogenic fibrosis [8] and formation of gadolinium deposits in the brain and body tissues [9,10,11]. This demonstrates that solutions allowing avoiding gadolinium administration are important.

Diffusion-weighted imaging (DWI) has shown a potential to replace contrast medium administration and to detect lesions before they come visible in conventional images [12]. DWI images use diffusion gradient applied in three perpendicular axes, with several b values (for example, 0, 50, 800 s/mm^2^) [13]. Inflamed segments as high signal intensity (SI) zones are best identified in the DWI tracking images of high b value, most commonly, b = 800 s/mm^2^ [14]. Diffusion is measured quantitatively by the apparent diffusion coefficient (ADC). Nevertheless, in spite of the high sensitivity and specificity of adding DWI to the MR imaging protocol, the specificity of DWI alone in assessment of IBD is still low, being 39–61% [15,16], since intact bowel walls also often present high SI in DWI tracking images of high b values (Figure 1).

A derivation of DWI, diffusion-weighted imaging with background body signal suppression (DWIBS), was introduced for whole body imaging of patients in order to detect metastases and tumour relapse. DWIBS provides more uniform fat suppression through the use of short T1 inversion recovery (STIR) based approach. This is a free breathing technique permitting multiple number of signal acquisitions to average motion [17]. Therefore, DWIBS provides a higher contrast-to-noise ratio, reduced image distortion, and better detection of subtle lesions [18]. A disadvantage of DWIBS is poorer signal-to noise ratio (SNR) [19], resulting in grainy image appearance. Despite the extensive use of DWIBS, there are very few studies regarding its use in assessment of the digestive tract. Tomizawa et al. performed studies on DWIBS in regards to assessing gall bladder walls and gastrointestinal tract [20,21,22,23]. Several researchers included a free breathing DWI sequence in their MRE protocol for the assessment of inflammatory bowel disease [13,24,25], however, no research data exist on comparison between DWI and DWIBS in bowel imaging. Whilst similar to other body tissue, the bowel wall presents a better resolution in DWIBS, compared to DWI sequence in both intact (Figure 1) and inflamed (Figure 2) bowel walls. Besides, DWIBS is more reproducible than DWI [26], and ADC-DWIBS values are also unaltered by motion [27]. Therefore, DWIBS could be beneficial over routinely used DWI in assessment of bowel inflammation.

The purpose of our study was to estimate the potential significance of using ADC measurements in DWI and DWIBS in patients, without preparing bowels with a hyperosmolar enteric contrast agent by setting the following tasks:assessing ADC-DWI and ADC-DWIBS values in intestines without preparation (collapsed bowel) and after preparation with hyperosmolar enteric contrast agent (filled bowel);assessing ADC-DWI and ADC-DWIBS values in the colon without preparation (in presence of intraluminal faeces) and after ingestion of enteric contrast agent (presence of it in the bowel lumen);comparing the consistency between ADC values of DWI and DWIBS, in conditions with and without bowel preparation in both intestines and the colon, and analyzing the utilization of DWIBS in MR of bowel imaging.

## 2. Materials and Methods

### 2.1. Patient Population

This prospective study included 106 primary care patients (18–76 years old), referred to MRE from March 2015 until March 2018, due to dyspeptic complaints but with no clinical and/or morphological evidence of IBD. The inclusion criteria were: absence of typical IBD symptoms—diarrhea, bloody and/or mucous stool, severe and/or crampy abdominal pain and rectal involvement [28,29].

The exclusion criteria were: age <18 years, fecal calprotectin (FC) level >200 μg/g, acute bowel infection, proven or previously diagnosed IBD, endoscopically proven enteropathy (e.g., coeliac disease, collagenous colitis etc.), present bowel tumor, and systemic diseases such as cystic fibrosis.

### 2.2. The Study

In this prospective observational cross-sectional study, ADC of DWI and DWIBS were assessed in bowel walls before and after preparation with hyperosmolar enteric contrast agent. Prior to bowel preparation, DWI and DWIBS scanning sequences were performed in all patients. Afterwards, patients were given enteric contrast agent, and scanned as per complete MRE protocol with DWI and DWIBS sequences included.

The study included two cohorts: (1) assessment of ADC-DWI and ADC-DWIBS intestinal walls before and after patient preparation, and (2) assessment of ADC-DWI and ADC-DWIBS in colonic walls before and after preparation. ADC measurements were only performed on bowel segments where high SI was present in DWI tracking images of b = 800 s/mm^2^. In order to compare ADC before and after bowel preparation, only patients with measurements both before and after preparation were included in the further data analysis. Similarly, for comparison between ADC-DWI and ADC-DWIBS, only patients with measurements performed in both DWI and DWIBS sequences were included in the further analysis. In both cohorts, data were grouped by the preparation state of the patient—non-prepared versus prepared bowels—and mutually compared.

All patients fasted for at least 6 h prior to MRE. After the initial scanning of DWI and DWIBS in prone position, patients were instructed to intake 1.250–1.500 l of 2.5% mannitol solution within 45–60 min, followed by full MRE exam in prone position. During the MRE exam and prior to DWI and DWIBS sequences, 20 mg dose of butylscopolamin was intravenously administered to reduce bowel peristalsis.

Patients were scanned with 1.5 T MRI system (Ingenia, Philips Medical Systems, Best, The Netherlands) using a 16-channel body coil. The applied DWI and DWIBS protocols were obtained from the Philips standard abdominal protocol and included in the protocol repository of the MRI system. To enable DWIBS-ADC measurements, the standard DWIBS protocol was amended by replacing a single b factor b = 1000 s/mm^2^ by three b factors 0 s/mm^2^, 600 s/mm^2^ and 800 s/mm^2^, consistent to DWI protocol.

The scanning parameters for DWI and DWIBS protocols are given in the Table 1.

The study was approved by the Ethics committee of Riga Stradin’s University, and written informed consent was obtained from all patients. The permission number by the Ethics committee of Riga Stradin’s University is 6/10.09.2015.

### 2.3. 1st Cohort

The first cohort was formed of patients in whom high SI bowel walls in DWI tracking images of b = 800 s/mm^2^ were identified in at least one intestinal site. High SI regions in at least one intestinal region were identified in all 106 patients. Prior to bowel preparation, one collapsed jejunal segment in DWI and DWIBS image series was identified for each patient. After bowel preparation, one distended jejunal segment in DWI and DWIBS image series was identified for each patient. ADC values were measured in three sites per segment using 10–20 mm^2^ oval region of interest (ROI), both before and after preparation (Figure 3).

### 2.4. 2nd Cohort

The second cohort was formed of patients in whom high SI bowel walls in DWI tracking images of b = 800 s/mm^2^ were identified in at least one colonic site. 78 of the 106 patients were identified to have high SI regions in at least one colonic region. Before bowel preparation in the DWI and DWIBS image series, one caecum or ascending colon segment with presence of intraluminal faeces was identified in each patient. After bowel preparation in the DWI and DWIBS image series, one caecum or ascending colon segment with presence of intraluminal mannitol was identified in each patient. ADC values were measured in three sites per segment using 10–20 mm^2^ ROI both before and after preparation (Figure 4).

### 2.5. Statistical Analysis

Statistical analysis was performed using software Stata/IC (StataCorp LLC, Texas, USA), mean ADC values were compared with paired *t*-test, and 99% confidence intervals (CI) were calculated for differences. The statistical significance of differences between mean values within groups was determined using one-way ANOVA with Bonferroni correction. *P* value of <0.05 was considered to be statistically significant.

## 3. Results

### 3.1. 1st Cohort: Comparison of ADC-DWI and ADC-DWIBS Measurements in Non-Prepared and Prepared Intestines

To perform the ADC measurements amongst the 106 patients, ADC-DWI was measured in 91 collapsed and 106 distended jejunal segments. ADC-DWIBS was measured in 86 collapsed and in 95 distended jejunal segments. Comparisons were drawn by analysis of segment pairs before and after preparation as follows; 88 segment pairs for comparison between ADC-DWI and 83 pairs for ADC-DWIBS. For comparison between ADC-DWI and ADC-DWIBS before preparation 85 segment pairs were analyzed, and 95 pairs were analyzed to compare ADC-DWI and ADC-DWIBS after preparation.

In both DWI and DWIBS sequences, the study found marked significant difference between ADC of non-prepared and prepared bowels. In both DWI and DWIBS ADC, the values of non-prepared jejunum were lower than in prepared jejunum. The ADC difference between non-prepared and prepared bowel was 38.1% in DWI and 48% in DWIBS. The ADC values are shown in Table 2.

Within the walls of non-prepared jejunum, our data showed a statistically significant ADC difference (*p* < 0.0001) of 16.5% between DWI and DWIBS, being lower in DWIBS. No significant ADC difference (*p* = 0.84) between DWI and DWIBS was observed within walls of prepared jejunum.

### 3.2. 2nd Cohort: Comparison of ADC-DWI and ADC-DWIBS Measurements in Non-Prepared and Prepared Intestines

To perform the ADC measurements amongst the 106 patients, ADC-DWI was measured in 41 non-prepared and in 42 prepared caecum or ascending colon segments. ADC-DWIBS was measured in 25 non-prepared caecum or ascending colon segments and in 18 prepared caecum or ascending colon segments. Forty-one segment pairs were analyzed for comparison between ADC-DWI before and after preparation, and 18 segment pairs were analyzed to compare ADC-DWIBS before and after preparation. For comparison between ADC-DWI and ADC-DWIBS before preparation, 25 segment pairs were analyzed, and 18 pairs were analyzed to compare ADC-DWI and ADC-DWIBS after preparation.

In both DWI and DWIBS sequences, the study found marked significant difference between ADC in non-prepared and prepared bowels. In both DWI and DWIBS ADC, values of the non-prepared colon were lower than in the prepared colon. The ADC difference between non-prepared and prepared bowel was 33.8% in DWI and 50.5% in DWIBS. The ADC values are presented in Table 3.

By mutually comparing ADC-DWI and ADC-DWIBS values within walls of both non-prepared and prepared colon, the data showed statistically significant ADC difference (*p* < 0.0001) of 28.4% between DWI and DWIBS being lower in DWIBS. No significant ADC difference (*p* = 0.58) between DWI and DWIBS values was found.

## 4. Discussion

According to the current joint evidence-based guidelines by the European Chron’s and Colitis Organization as well as European Society of Gastrointestinal and Abdominal Radiologists, MRE imaging in IBD requires administration of contrast medium giving opportunity to estimate the bowel wall enhancement pattern [5] as well as vasa recta engorgement commonly seen in CD [30]. A significant advantage of using the gadolinium contrast agent is the ability to quantify Crohn’s disease activity [31]. Nevertheless, according to literature data, DWI in detection of bowel inflammatory changes outperforms T1 dynamic series with intravenous gadolinium contrast agent [32,33]. Therefore, DWI could be beneficial in case of diagnostic difficulties, for example, when, in Crohn’s disease, the inflamed bowel tissues are covered by intact mucosa [34]. The drawback of DWI is its low specificity [15,16].

In intact bowel walls, high intensity signal resembling inflammation in images of high b factors is a reason for the low specificity of DWI. This pattern is commonly explained with the T2 shine-through effect, where theoretically ADC should be high [35]. Nevertheless, increased DWI signal along with low ADC is observed not only in inflamed but also in disease-free bowel walls [36].

Upon reporting multiple MRE exams, we had several observations regarding the high SI bowel wall at the DWI tracking images of b = 800 s/mm^2^. Firstly, we noticed that intestinal SI was markedly higher in bowel wall before preparation, i.e., in totally collapsed bowel, compared to intestinal wall after preparation, i.e., in fully distended bowel. Secondly, in colon, SI was markedly higher in bowel wall after preparation, i.e., in presence of enteric contrast agent, compared to the colonic wall before preparation, i.e., in presence of faeces. In both situations, high SI bowel walls in the ADC map frequently presented low SI. These observations raised a question regarding ADC differences between bowel wall before and after patient preparation, in both intestines and colon.

Results from the 1st cohort comparing ADC of DWI and DWIBS between non-prepared (collapsed) and prepared (distended) intestines showed that ADC values in both DWI and DWIBS in the collapsed bowel sample were markedly lower than in distended bowel samples. As the bowel collapses, the number of cells per volume unit increases; however, the cells itself are not altered. Therefore, the volumes of intracellular and extracellular spaces were still constant, giving no reason for restricted diffusion. The measurement results could be explained by the partial volume effect. In prepared (filled) jejunal wall, the signal of very thin bowel wall was contaminated by the high intensity signal from the massive volume of enteric contrast agent, therefore, ADC value is high. However, in non-prepared (collapsed) bowel samples, the amount of high SI intraluminal content is less, therefore, the contamination of the intestinal wall signal is also less.

A similar explanation applies to the 2nd cohort comparing ADC of DWI and DWIBS between non-prepared (presence of low SI intraluminal faeces) and prepared (presence of mannitol) colon. The results showed that ADC values in both DWI and DWIBS were dependent on colonic intraluminal content and in the presence of low signal intensity faeces were nearly two times lower than in the presence of high signal intensity mannitol.

According to a number of studies, the performance of DWI varied among authors. The range of ADC values in normal bowel wall was 1.18–3.69 mm^2^/s whereas in inflamed bowel segments—1.24–1.988 mm^2^/s being significantly lower for 0.8–2.4 ×10^−3^ mm^2^/s than in intact bowels. Several authors have also provided their cut-off ADC values for discriminating between inflamed and intact bowel walls. These values are mutually different and lie between ADC ranges of inflamed and intact bowel walls, except in one study where cut-off value lies within the range of the inflamed bowel. According to data from all researchers, ADC ranges of IBD and intact bowel do not mutually overlap [37]. Most of these studies have been performed in prepared bowels. To the best of our knowledge, a team of Kiryu et al. is the only research group reporting ADC values in Crohn’s disease patients without patient preparation using free-breathing DWI (i.e., using STIR as fat suppression method). The reported ADC values show a similar trend, with that in prepared bowels being lower in disease-active segments and higher in disease-inactive areas (1.61 ± 0.44 × 10^−3^ mm^2^/s versus 2.56 ± 0.51×10^−3^ mm^2^/s in intestines, respectively) [13]. This difference is high enough to concern the potential benefit of ADC measurements without bowel preparation. If the consistency of differences between ADC values of inflamed and normal bowels applies also in non-prepared bowel samples, this would allow a proper estimation of ADC in patients without preparation. Therefore, assessment of consistency between ADC values of non-prepared and prepared bowel walls was a goal of our research.

Comparing the obtained ADC values of bowel prior preparation and prepared bowel to literature data on ADC values of normal and inflamed bowels, it is obvious that, although the ADC values for the prepared bowels are markedly higher than ADC values in bowel prior to preparation, both ADC values of prepared and non-prepared bowel loops overlap with the ADC range of inflamed bowel provided in literature [37]. Thus, the applicability of ADC in non-prepared bowel samples is still questionable and might be related to scanning conditions and parameters in our institution. to fully compare the extent of overlap in ADC ranges for non-prepared bowel and inflamed bowels, the ADC values for CD have to be obtained in our institution under identical conditions.

A questionable issue is how much mannitol itself changes ADC, by impacting cells due to concentration gradient between bowel epithelium cells of lower and higher osmolarity substance, and the expelling fluid from cells resulting in a lowered packed cell volume [38] and osmotic diarrhoea [6]. Nevertheless, we assume that as the bowel wall filled with mannitol is a very thin (<3 mm), the contribution to signal intensity by osmotic influence of mannitol is negligible.

By mutually comparing ADC-DWI and ADC-DWIBS, we observe no statistically significant difference in the wall of prepared bowels, both regarding intestines and the colon. On the contrary, ADC values in non-prepared bowels and both intestines and colon, is markedly lower than in the prepared bowel samples. In the prepared bowel, the wall signal was influenced with a high large amount of fluid. In the non-prepared bowel, there was a presence of high-viscosity intraluminal content—chime in intestines and faeces in colon. DWI and DWIBS sequences differed when fat suppression techniques were used. In DWI, CHESS (CHEmically Selective Saturation) or SPIR sequences suppress fat selectively. The STIR technique used in DWIBS was based on the T1 relaxation time of tissues [39], and suppressed signals from all substances of short T1 values such as proteinaceous, viscous and mucous substances, including chime and faeces, thus being non-selective [40]. Therefore, ADC-DWIBS values are lower in the presence of the bowel content, comparing to ADC-DWI, and the probability that the ADC-DWIBS range will overlap the ADC range of inflamed bowel is higher. Therefore, based on the findings of this study, we still recommend preferring SPIR-based ADC measurements and using ADC values of STIR-based ADC in non-prepared bowel with caution.

Our study had several limitations. 1) Measurements were performed by one radiologist not assessing inter-observer agreement, and in such a small volume ADC values are reported to be hardly reproducible [37,41] as they rest on subjectivity. Nevertheless, data from ADC measurements in liver imaging suggested better reproducibility of free-breathing DWIBS over respiratory-triggered DWI [26], which could be also proven better for bowel walls, but requires further investigation. 2) Typically, achievable DWI resolution is on the order of 2 × 2 × 2 mm^3^ [42]. The pixel size used in our standard protocols (see Table 1) could be large for tiny structures, such as the bowel wall. ADC values are therefore markedly impacted by the partial volume effect being very approximate and can be used only for reference but not as absolute values. 3) In intestines, the most uniform luminal distension was present in jejunum, which was therefore chosen for intestinal measurements, however, ileum is the main location of CD. The terminal loop of ileum also has different morphological patterns—abundance of lymphoid tissues [43]—which also could influence ADC measurements. In the colon, measurements were performed only in walls of the caecum and the ascending colon, since presence or mannitol was mainly observed in these locations. 4) Location of the sites with high SI signal in DWI tracking images of b = 800 s/mm^2^ was not consistent among the series, therefore, measurements could not be performed precisely at the same locations. 5) We did not pay special attention to the T2 shine through effect of bowel walls, and measured ADC values in DWI tracking images of b = 800 s/mm^2^ regardless of signal appearance in ADC map. 6) Our goal was to observe properties of DWI-ADC and DWIBS-ADC in sites of bowel walls showing high SI at DWI tracking images of b = 800 s/mm^2^ resembling bowel inflammation, whereas we did not consider other signs of bowel inflammation like oedema, increased bowel wall thickness, contrast enhancement, etc., which of course were absent in patients with no presence of IBD as required by the study.

## 5. Conclusions

The study has found that ADC ranges of the non-prepared bowel are significantly lower when compared to the ADC ranges of prepared bowels in both intestines and the colon, potentially overlapping ADC ranges of the inflamed bowel in both DWI and DWIBS sequences. Since ADC values could be related to scanning conditions at a particular institution, ADC values of the intact bowel have to be compared to ADC values obtained from CD patients at our institution, scanned with the same equipment and under identical scanning parameters and conditions.

In prepared bowels, ADC-DWIBS and ADC-DWI measurements are equally useful for bowel wall measurements. In non-prepared bowels, ADC-DWIBS could be disadvantageous over ADC-DWI in revealing IBD, as ADC-DWIBS values are markedly lower than ADC-DWI values, and they potentially overlap ADC ranges of IBD to a greater extent than ADC-DWI.

## Figures and Tables

**Figure 1 medicina-55-00394-f001:**
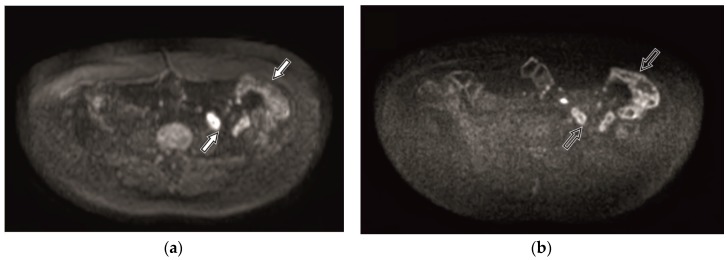
Diffusion-weighted imaging (DWI) (**a**) and diffusion-weighted imaging with background body signal suppression (DWIBS) (**b**) tracking images of b = 800 s/mm^2^ within the magnetic resonance enterography (MRE) of 41-year-old female patient with intact bowels. The unaltered bowel wall shows high signal intensity (SI). Better resolution of the bowel loop in the DWIBS image (**b**, black arrow) is seen comparing to the DWI image (**a**, white arrow).

**Figure 2 medicina-55-00394-f002:**
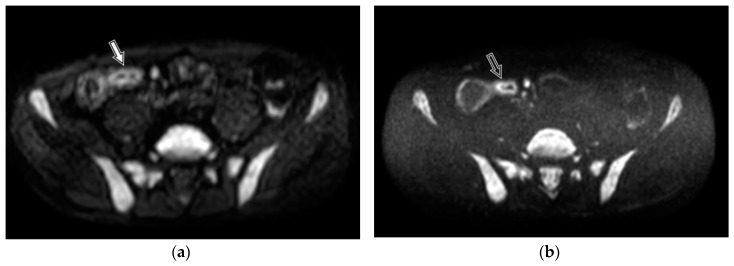
DWI (**a**) and DWIBS (**b**) tracking images of b = 800 s/mm^2^ within the MRE of 40-year-old male patient with histologically proven Crohn’s disease. Altered bowel loop shows high SI. Better resolution and delineation of the inflamed bowel mucosa in the DWIBS image (**b**, black arrow) is seen as compared to the DWI image (**a**, white arrow).

**Figure 3 medicina-55-00394-f003:**
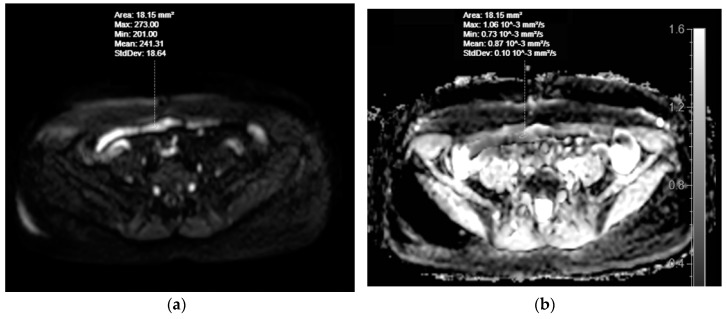
MRE of a 53-year-old female patient with dyspeptic complaints, with no morphologically verified IBD. Selecting regions of interest (ROIs) for apparent diffusion coefficient (ADC) measurements in DWI images of collapsed (**a**) and distended (**c**) jejunum showing high SI in the DWI tracking images of b = 800 s/mm^2^. ADC values appear on the ADC map (**b**, for collapsed jejunum, **d**, for distended jejunum).

**Figure 4 medicina-55-00394-f004:**
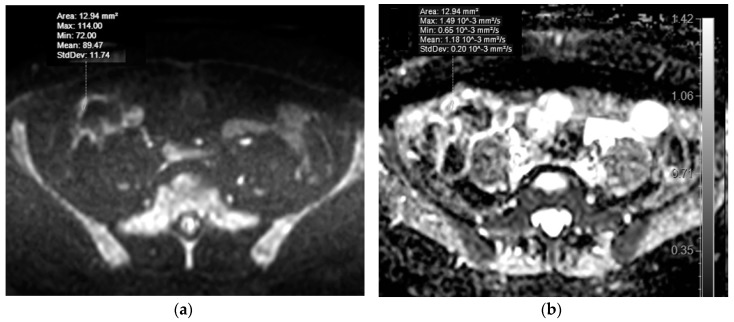
MRE of a 45-year-old male patient with dyspeptic complaints, with no morphologically verified inflammatory bowel disease (IBD). Selecting ROIs for ADC measurement in DWIBS tracking images (b = 800 s/mm^2^) of the ascending colon wall before preparation of patient with mannitol, at presence of intraluminal faeces **(a)** and after preparation, at presence of enteric contrast agent **(b)**, showing high SI. Note the higher SI of colonic wall in presence of mannitol, when compared to the presence of faeces in the colonic lumen. ADC values appear on the ADC map (**b**, for non-prepared colon, **d**, for prepared colon).

**Table 1 medicina-55-00394-t001:** Scanning parameters for DWI and DWIBS protocols.

Scanning Protocol	DWI ^1^	DWIBS ^2^
Sequence	SE-EPI ^3^	STIR-EPI ^4^
Mode	Single shot	Single shot
Coil	SENSE ^5^ body	SENSE body
Slice orientation	Axial	Axial
FOV ^6^	RL ^7^ 400 mm, AP ^8^ 350 mm, FH ^9^ 303 mm	RL 400 mm, AP 350 mm, FH 303 mm
ACQ ^10^ voxel size	RL 3.03 mm x AP 3.57 mm x slice thickness 6 mm	RL 2.50 mm x AP 2.98 mm x slice thickness 6 mm
Reconstruction voxel size	RL 1.79 mm x AP 1.79 mm x slice thickness 6 mm	RL 1.39 mm x AP 1.39 mm x slice thickness 6 mm
Fold-over suppression	No	No
Reconstruction matrix	224	288
SENSE	Yes	Yes
P reduction (AP)	2	2.5
Number of stacks	1	1
Type	Parallel	Parallel
Slices	46	46
Slice gap (mm)	0.6	0.6
Slice orientation	Transverse	Transverse
Fold-over direction	AP	AP
Fat shift direction	A	P
TE ^11^	66 ms	78 ms
TR ^12^	1426 ms	7055 ms
TI ^13^	-	180 ms
Fast imaging mode	EPI ^14^	EPI
Flip angle	90°	
Fat suppression	SPIR ^15^	STIR ^16^
b factors	0, 600, 800	0, 600, 800
Respiratory compensation	Trigger	No
Number of signal averages	3	5
Acquisition time	4 min 12 s	5 min 56 s

^1^ DWI, diffusion-weighted imaging; ^2^ DWIBS, diffusion-weighted imaging with background body signal suppression; ^3^ SE-EPI, Spin-Echo – Echo Planar Imaging; ^4^ STIR-EPI, Short T1 Inversion Recovery-Echo Planar Imaging; ^5^ SENSE, SENSitivity Encoding; ^6^ FOV, Field of View; ^7^ RL, Right-Left direction; ^8^ AP, Anterior-Posterior direction; ^9^ FH, Foot-Head direction; ^10^ ACQ, Acquisition; ^11^ TE; ^12^ TR, Repetition Time; ^13^ TI, Inversion Time; ^14^ EPI, Echo Planar Imaging; ^15^ SPIR, Spectral Presaturation with Inversion Recovery; ^16^ STIR, short T1 inversion recovery.

**Table 2 medicina-55-00394-t002:** Comparison of ADC values between DWI and DWIBS in walls of non-prepared and prepared jejunum.

Bowel Preparation State	Non-Prepared (Collapsed) Jejunum	Prepared (Filled) Jejunum	*p* Value	Difference between Mean ADC Values of Filled vs. Collapsed Loops
Mean ADC-DWI value ×10^−3^ mm^2^/s	1.09 (SD = 0.37)	1.76 (SD = 0.41)	<0.0001	0.67 (38.1%)
Mean ADC-DWIBS value ×10^−3^ mm^2^/s	0.91 (SD = 0.47)	1.75 (SD = 0.51)	<0.0001	0.84 (48%)

ADC, apparent diffusion coefficient; SD, standard deviation.

**Table 3 medicina-55-00394-t003:** Comparison of ADC values between DWI and DWIBS in walls of non-prepared and prepared colon.

Bowel Preparation State	Non-Prepared (Presence of Faeces) Colon	Prepared (Presence of Mannitol) Colon	*p* Value	Difference between Mean Values in Non-Prepared vs. Prepared Colon
Mean ADC-DWI value × 10^−3^ mm^2^/s	1.41 (SD = 0.31)	2.13 (SD = 0.41)	<0.0001	0.72 (33.8%)
Mean ADC-DWIBS value × 10^−3^ mm^2^/s	1.01 (SD = 0.40)	2.04 (SD = 0.58)	<0.0001	1.03 (50.5%)

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
