# Peer review of "The Influence of Bowel Preparation on ADC Measurements: Comparison between Conventional DWI and DWIBS Sequences"

_1010-660X, 2019, doi:10.3390/medicina55070394_

Round 1

Reviewer 1 Report

In their manuscript "The influence of bowel preparation on DWI-based ADC measurements: comparison between conventional DWI and DWIBS sequences" (medicina-531929), the authors evaluate  ADC measurements of the bowel walls with and without bowel preparation as well as with two different pulse sequences (conventional EPI-DWI sequence and DWI with background signal suppression (DWIBS) in 106 patients without evidence for inflammatory bowel disease (IBD).

They find signficant ADC differences; in particular, ADCs after bowel preparation were signficantly higher. The authors hypothesize that the observed ADC differences results from partial volume effects from intrabowel signal.

This is a generally well performed study with interesting results and conclusions. A relevant limitation is that the measurements were performed only by one radiologist; it would be very interesting to learn about inter-observer agreement, since the evaluated ROIs are very small. At least, the influence of the small ROIs and the problematic positioning should be discussed in more detail; e.g. can the partial volume effects be assessed quantitatively by evaluating the standard deviations or maximum values of the ROIs?

Specific comments:

Title: remove "DWI-based". Check if ADC, DWI, and DWIBS should be written out in the title.

Abstract: please add correct units to all b-values (here and in the main text and in the figure captions and table titles): "s/mm²"!    

Introduction: p.2,l.64: What is meant by "signal-to-contrast ratio"? Do you mean "contrast-to-noise ratio"?

Material and Methods: p.3,l.113: "were solely performed": unclear wording; maybe, you should move "Prior to bowel preparation," at the beginning of this sentence and use "only" instead of "solely"?

p.4,l.134: Please relate the ROI size of 10-20 mm² to the number of pixels (real resolution and interpolated).

p.4,l.134: Please clarify if exactly the same ROI was used on the DWI images and the corresponding ADC maps. Were there ROIs copied?

p.3/6: In my opinion, the Methods section would be clearer if the "MRI examination" and "Image analysis" subsections would be moved in front of the "Study" section.

p.6, Table 1: Please add the total scan time of both pulse sequences (whould be substantially longer for the DWIBS sequence due to longer TR and higher number of averages).

Discussion: p.9,l.239: please add "signal" at: "increased DWI _signal_ along ..."

p.9,l.252: "increased number of cells per volume ... volumes of intracellular and extracellular spaces were still constant" - this does not sound right. More cells per volume should mean less space per cell. Please check.

References: Please correct the numbering (starts with 2).

Author Response

Dear Sir/Madam,

Thank you very much for the positive words, suggestions and comments on our research, as well as the questions helping us to improve the article.

Please find our  responses below, following your concerns and questions.

Your comment: A relevant limitation is that the measurements were performed only by one radiologist; it would be very interesting to learn about inter-observer agreement, since the evaluated ROIs are very small. At least, the influence of the small ROIs and the problematic positioning should be discussed in more detail; e.g. can the partial volume effects be assessed quantitatively by evaluating the standard deviations or maximum values of the ROIs?

We agree that the lack of inter- or intra-observer agreement is a relevant limitation of the study. In this research, we tried to place the ROI symmetrically in relation to the inner and outer contours of the gut therefore, with regard to the standard deviation, we believe that the measurements were performed to obtain the lowest standard deviation possible in DWI. Of course, the reliability of measurements would be depicted better by assessing the inter- or intra-observer agreement. Therefore, we improved the consecutive research regarding ADC-DWI and ADC-DWIBS measurements in patients with active Crohn’s disease performing two measurement sets for assessment of intra-observer agreement. The results are still undergoing statistical processing.

Your comment: Title: remove "DWI-based". Check if ADC, DWI, and DWIBS should be written out in the title.

“DWI-based” removed. The title now is “The influence of bowel preparation on ADC measurements: comparison between conventional DWI and DWIBS sequences”.

Your comment: Abstract: please add correct units to all b-values (here and in the main text and in the figure captions and table titles): "s/mm²"! 

Changes done.

Your comment: Introduction: p.2,l.64: What is meant by "signal-to-contrast ratio"? Do you mean "contrast-to-noise ratio"?

Yes, contrast-to-noise ratio. It was a mistyping error. Thank you for awareness.

Your comment: Material and Methods: p.3,l.113: "were solely performed": unclear wording; maybe, you should move "Prior to bowel preparation," at the beginning of this sentence and use "only" instead of "solely"?

Thank you for the style suggestions. Changes done.

Your comment: p.4,l.134: Please relate the ROI size of 10-20 mm² to the number of pixels (real resolution and interpolated).

For DWI, the real pixel size is 10.81 mm2 (meaning that the ROI contains 0.93-1.85 pixels), the interpolated pixel size – 3.20 mm2(meaning that the ROI contains 3.1-6.2 pixels). For DWIBS, the real pixel size is 7.45 mm2 (meaning that the ROI contains 1.32-2.68 pixels) and the interpolated pixel size – 1.93 mm2 (meaning that the ROI contains 5.26-10.53 pixels). Since an intact bowel wall is very thin (up to 3 mm) a single pixel in DWI (both real and interpolated one) covers the anatomy outside its border being the reason for the partial volume effect in more extent comparing to DWIBS where theoretically the pixel could lie within the borders of the thin bowel wall. Nevertheless, the SD for ADC-DWIBS exceeds the SD of ADC-DWI for 0.1-0.17 which probably could be related to the markedly lower signal-to-noise ratio of DWIBS comparing to DWI.

In the text we already noted that the pixel size used could be too large for the tiny structures as the bowel wall is. Since the smaller pixel size in DWIBS does not benefit the dispersion of the ADC values in terms of SD, we decided not to detail the abovementioned concern in the text.

Your comment: p.4,l.134: Please clarify if exactly the same ROI was used on the DWI images and the corresponding ADC maps. Were there ROIs copied?

In the postprocessing server Philips Intellispace we used for image analysis as the ROI is chosen on the DWI tracking images it appears automatically on the ADC map therefore the ROI appearing on the DWI tracking images and the corresponding ADC map was exactly one and the same. In the description of limitations it was said that the ROIs were put in the DWI images of b=800 s/mm2(that means, not on the AC map). Probably it was not clear enough therefore I added in the text that the ROIs were chosen in the DWI tracking images of b=800 s/mm2.

Your comment: p.3/6: In my opinion, the Methods section would be clearer if the "MRI examination" and "Image analysis" subsections would be moved in front of the "Study" section.

Thank you for the suggestion. Corrections done.

Your comment: p.6, Table 1: Please add the total scan time of both pulse sequences (whould be substantially longer for the DWIBS sequence due to longer TR and higher number of averages).

The total scan time for DWI sequence is 4 min 12 sec, for DWIBS – 5 min 56 sec. The table is updated.

Your comment: Discussion: p.9,l.239: please add "signal" at: "increased DWI _signal_ along ..."

Response: Thank you, changes made.

Your comment: p.9,l.252: "increased number of cells per volume ... volumes of intracellular and extracellular spaces were still constant" - this does not sound right. More cells per volume should mean less space per cell. Please check.

What we meant was the something different. When a bowel loop is distended, the number of cells per volume unit, e.g., 1 mm3 is low since the major part of this volume is filled mainly with bowel content and the amount of cells is less. If the bowel is collapsed, the microvilli of the bowel wall collapse like accordion, the cells come closer together. As a result, the amount of bowel content – fluid of faeces, is relatively less, but the number of bowel cells per this volume unit increases. However, the volumes of the relevant intracellular – extracellular spaces remain the same. I made some amendments in the text making this idea more clear.

Your comment: References: Please correct the numbering (starts with 2).

Response: Corrections done.

Reviewer 2 Report

Summary:

The authors compared the apparent diffusion coefficient (ADC) values or bowels prepared with an enteric contrast agent using two sequences: conventional diffusion weighted imaging (DWI), and diffusion weighted imaging with background suppression (DWIBS). 

The authors report that (a) the measured ADC values after bowel preparation were significantly higher than without bowel preparation for either technique, and (b) the ADC values measured using either technique were equivalent.

The conclusions of the paper with respect to increase in the measured ADC post bowel preparation compared to that measured without, and that  ADC values obtained with either conventional DWI or DWIBS are equivalent, is well supported by the evidence presented by the authors. 

Crtique:

This paper could benefit from copy editing to make it a bit more concise and clear.  I have identified a few items below:

line 50: Perhaps, the authors should revise the sentence as: "Inflamed segments are identified as regions of elevated signal intensity in high 'b' value DWI ( b>= 800 s/mm2).

line 64:  Here authors state that "DWIBS provides a higher signal-to-contrast ratio", and it is unclear what is signal-to-contrast ratio.  The last sentence in the same paragraph also states:"... DWIBS is poorer signal-to-noise ratio (SNR)...". 

I recommend that the long description of DWIBS be abbreviated to one or two sentences highlighting that DWIBS provides uniform fat suppression through the use of an inversion-recovery based approach, and that it is a free breathing technique - that permits multiple NSA to average motion.

line 88-110: Shorten the purpose.  Perhaps, combining goals 1 and 2 into a single statement, e.g., comparing the ADC values of DWI of the intentines and colon obtained using ADC-DWI and ADC-DWIBS, before and after preparation, could be more succinct.

Tables 2 and 3 could be combined, and the results could also be presented as one set of values for jejunum and colon- before and after preparation, and with two techniques.

In the discussion, briefly comment on any added value of  contrast-enhanced MRI versus DWI based techniques for the evaluation of CD. 

Furthermore, you acquired two 'b' values one at 600 s/mm2 and another at 800 s/mm2.  Comment on the utility of 600 s/mm2 acquisition. (Provide units for 'b' value whenever referenced).

Include one or two lines in the discussion, about what would be the recommended practice based on the findings of this study, and based on the literature (ref: 43, Dohan et al. JMRI, 2016;44:1381-1396).  This will help the reader understand this manuscript in the context of current literature.

Author Response

Please see in the attachment.

Reviewer 3 Report

This was an interesting prospective study with a great sample size. It could be potentially beneficial and have a great impact on clinical practice. However, here are several questions and comments.

1.     You described how you identified the high SI region in intestinal/colonic regions before preparation. But it is not clear how you identified the same location after the bowel preparation. It is required that you measure the wall ADC values at the same location for comparison.

2.     Reproducibility and repeatability of the measurements are not clearly done nor described. You did not report whether there is any inter- or intrareader agreement.

3.     Some statements in conclusion for example “DC-DWIBS has no advantage ADC-DWI in regard to IBD assessment” is overstated and beyond the scope of this analysis.

4.     Statistical tests being used seem not ideal for this analysis. Since authors are comparing patients before and after preparation, a paired comparison is required.

Author Response

Please see in the attachment.

Round 2

Reviewer 3 Report

You have made some of the requested modifications reasonably and satisfactorily. You have mentioned the limitation of your study that it is done by a single radiologist, which is not ideal but fine. In general, it is much better now.